# Sensitivity analysis with respect to observations in variational data assimilation for parameter estimation

Victor Shutyaev[1,2,3], Francois-Xavier Le Dimet[4], and Eugene Parmuzin[1,3]

[1]Institute of Numerical Mathematics, Russian Academy of Sciences, Gubkina 8, Moscow 119333 , Russia
[2]Federal State Budget Scientific Institution "Marine Hydrophysical Institute of RAS", Kapitanskaya, 2, Sevastopol 299011, Russia
[3]Moscow Institute for Physics and Technology, 9 Institutskiy per., Dolgoprudny, Moscow Region, 141701, Russia
[4]LJK, Université Grenoble Alpes, 700 Avenue Centrale, 38401 Domaine Universitaire de Saint-Martin-d'Hères, France

**Correspondence:** Victor Shutyaev (victor.shutyaev@mail.ru)

**Abstract.** The problem of variational data assimilation for a nonlinear evolution model is formulated as an optimal control problem to find unknown parameters of the model. The observation data, and hence the optimal solution, may contain uncertainties. A response function is considered as a functional of the optimal solution after assimilation. Based on the second-order adjoint techniques, the sensitivity of the response function to the observation data is studied. The gradient of the response function is related to the solution of a non-standard problem involving the coupled system of direct and adjoint equations. The non-standard problem is studied, based on the Hessian of the original cost function. An algorithm to compute the gradient of the response function with respect to observations is presented. Numerical example is given for variational data assimilation problem related to sea surface temperature for the Baltic Sea thermodynamics model.

## 1 Introduction

The methods of data assimilation (DA) have become an important tool for analysis of complex physical phenomena in various fields of science and technology. These methods allow us to combine mathematical models, data resulting from observations and a priori information. The problems of variational DA can be formulated as optimal control problems (e.g. Lions, 1968; Le Dimet and Talagrand, 1986) to find unknown model parameters such as initial and/or boundary conditions, right-hand sides in the model equations (forcing terms), distributed coefficients, based on minimization of the cost function related to observations. A necessary optimality condition reduces an optimal control problem to an optimality system which involves the model equations, the adjoint problem, and input data functions. The optimal solution depends on the observation data, and for future forecast it is very important to study the sensitivity of the optimal solution with respect to observation errors (Baker and Daley, 2000).

The necessary optimality condition is related to the gradient of the original cost function, thus to study the sensitivity of the optimal solution, one should differentiate the optimality system with respect to observations. In this case, we come to the so-called second-order adjoint problem (Le Dimet et al., 2002). The first studies of sensitivity of the response functions after assimilation with the use of second-order adjoint were done by Le Dimet et al. (1997) for variational data assimilation problem

aimed at restoration of initial condition, where sensitivity with respect to model parameters was considered. The equations of the forecast sensitivity to observations in a four-dimensional (4D-Var) data assimilation were derived by Daescu (2008). Based on these results, a practical computational approach was given by Cioaca et al. (2013) to quantify the effect of observations in 4D-Var data assimilation.

The issue of sensitivity is related to the statistical properties of the optimal solution (see Gejadze et al., 2008, 2011, 2013; Shutyaev et al., 2012). General sensitivity analysis in variational data assimilation with respect to observations for a nonlinear dynamic model was given by Shutyaev et al. (2017) to control the initial-value function. The dynamic formulation of the problem is important because it shows different implementation options (Gejadze et al., 2018).

    This paper is based on the results of Shutyaev et al. (2017) and presents the sensitivity analysis with respect to observations
in variational data assimilation aimed at restoration of unknown parameters of a dynamic model. We should mention the importance of the parameter estimation problem itself. A precise determination of the initial condition is very important in view of forecasting, however the use of variational data assimilation is not limited to operational forecasting. In many domains (e.g. hydrology) the uncertainty in the parameters is more crucial that the uncertainty in the initial condition (e.g. White at al., 2003). In some problems the quantity of interest can be represented directly by the estimated parameters as controls. For example,
in Agoshkov et al. (2015) the sea surface heat flux is estimated in order to understand its spatial and temporal variability. The problems of parameter estimation are common inverse problems considered in geophysics and in engineering applications (see Alifanov et al., 1996; Sun, 1994; Zhu and Navon, 1999; Storch et al., 2007). Last years an interest is rising to the parameter estimation using 4D-Var (Bocquet, 2012; Schirber at al., 2013; Yuepeng et al., 2018; Agoshkov and Sheloput, 2017).

    We consider a dynamic formulation of variational data assimilation problem for parameter estimation in a continuous form,
but the presented sensitivity analysis formulas with respect to observations do not follow from our previous results for the initial condition problem (Shutyaev et al., 2017) and constitute a novelty of this paper. Of course, the initial condition function may be also considered as a parameter, however, in our dynamic formulation we have two equations for the model: one equation for describing an evolution of the model operator (involving model parameters such as right-hand sides, coefficients, boundary conditions etc.), and another equation is considered as an initial condition.

This paper is organized as follows. In section 2, we give the statement of the variational DA problem for a nonlinear evolution model to estimate the model parameters. In Section 3, sensitivity of the response function after assimilation with respect to observations is studied, and its gradient is related to the solution of a non-standard problem. In Section 4 we derive an operator equation involving the Hessian to study the solvability of the non-standard problem, and give an algorithm to compute the gradient of the response function. A proof-of-concept analytic example with a simple model is given in Section 5
to demonstrate how the sensitivity analysis algorithm works. Section 6 presents an application of the theory to the DA problem for a sea thermodynamics model. Numerical examples are given in Section 7 for the Baltic Sea dynamics model. The main results are discussed in the Conclusions.

## 2  Statement of the problem

We consider the mathematical model of a physical process that is described by the evolution problem

$$
\begin{cases}
\dfrac{\partial \varphi}{\partial t} = F(\varphi, \lambda) + f, & t \in (0, T) \\
\varphi|_{t=0} = u,
\end{cases}
\tag{2.1}
$$

where the initial state $u$ belongs to a Hilbert space $X$, $\varphi = \varphi(t)$ is the unknown function belonging to $Y = L_2(0, T; X)$ with the norm $\|\varphi\|_Y = (\varphi, \varphi)_Y^{1/2} = (\int_0^T \|\varphi(t)\|_X^2 \, dt)^{1/2}$, $F$ is a nonlinear operator mapping $Y \times Y_p$ into $Y$, $Y_p$ is a Hilbert space (space of control parameters, or control space), $f \in Y$. Suppose that for given $u \in X, f \in Y$ and $\lambda \in Y_p$ there exists a unique solution $\varphi \in Y$ to (2.1) with $\dfrac{\partial \varphi}{\partial t} \in Y$. The function $\lambda$ is an unknown model parameter.

Let us introduce the cost function

$$
J(\lambda) = \frac{1}{2}(V_1(\lambda - \lambda_b), \lambda - \lambda_b)_{Y_p} + \frac{1}{2}(V_2(C\varphi - \varphi_{obs}), C\varphi - \varphi_{obs})_{Y_{obs}},
\tag{2.2}
$$

where $\lambda_b \in Y_p$ is a prior (background) function, $\varphi_{obs} \in Y_{obs}$ is a prescribed function (observational data), $Y_{obs}$ is a Hilbert space (observation space), $C : Y \to Y_{obs}$ is a linear bounded observation operator, $V_1 : Y_p \to Y_p$ and $V_2 : Y_{obs} \to Y_{obs}$ are symmetric positive definite bounded operators.

Let us consider the following data assimilation problem with the aim to estimate the parameter $\lambda$: for given $u \in X, f \in Y$, find $\lambda \in Y_p$ and $\varphi \in Y$ such that they satisfy (2.1), and on the set of solutions to (2.1), the functional $J(\lambda)$ takes the minimum value, i.e.

$$
\begin{cases}
\dfrac{\partial \varphi}{\partial t} = F(\varphi, \lambda) + f, & t \in (0, T) \\
\varphi|_{t=0} = u, \\
J(\lambda) = \displaystyle\inf_{v \in Y_p} J(v).
\end{cases}
\tag{2.3}
$$

We suppose that the solution of (2.3) exists. Let us note that the solvability of the parameter estimation problems (or identifiability) has been addressed, e.g., in Chavent (1983), Navon (1998). To derive the optimality system, we assume the solution $\varphi$ and the operator $F(\varphi, \lambda)$ in (2.1)–(2.2) are regular enough, and for $v \in Y_p$ find the gradient of the functional $J$ with respect to $\lambda$:

$$
J'(\lambda)v = (V_1(\lambda - \lambda_b), v)_{Y_p} + (V_2(C\varphi - \varphi_{obs}), C\phi)_{Y_{obs}} = (V_1(\lambda - \lambda_b), v)_{Y_p} + (C^* V_2(C\varphi - \varphi_{obs}), \phi)_Y,
\tag{2.4}
$$

where $\phi$ is the solution to the problem:

$$
\begin{cases}
\dfrac{\partial \phi}{\partial t} = F'_\varphi(\varphi, \lambda)\phi + F'_\lambda(\varphi, \lambda)v, \\
\phi|_{t=0} = 0.
\end{cases}
\tag{2.5}
$$

Here $F'_\varphi(\varphi, \lambda) : Y \to Y$, $F'_\lambda(\varphi, \lambda) : Y_p \to Y$ are the Fréchet derivatives of $F$ (Marchuk et al., 1996) with respect to $\varphi$ and $\lambda$, correspondingly, and $C^*$ is the adjoint operator to $C$ defined by $(C\varphi, \psi)_{Y_{obs}} = (\varphi, C^*\psi)_Y$, $\varphi \in Y, \psi \in Y_{obs}$.

Let us consider the adjoint operator $(F'_\varphi(\varphi,\lambda))^* : Y \to Y$ and introduce the adjoint problem:

$$\begin{cases} \dfrac{\partial \varphi^*}{\partial t} + (F'_\varphi(\varphi,\lambda))^* \varphi^* &= C^* V_2 (C\varphi - \varphi_{obs}), \\ \varphi^* \big|_{t=T} &= 0. \end{cases} \tag{2.6}$$

Then (2.4) with (2.5) and (2.6) gives

$$J'(\lambda)v = (V_1(\lambda - \lambda_b), v)_{Y_p} - (\varphi^*, F'_\lambda(\varphi,\lambda)v)_Y = (V_1(\lambda - \lambda_b), v)_{Y_p} - ((F'_\lambda(\varphi,\lambda))^* \varphi^*, v)_{Y_p}, \tag{2.7}$$

where $(F'_\lambda(\varphi,\lambda))^* : Y \to Y_p$ is the adjoint operator to $F'_\lambda(\varphi,\lambda)$. Therefore, the gradient of $J$ is defined by

$$J'(\lambda) = V_1(\lambda - \lambda_b) - (F'_\lambda(\varphi,\lambda))^* \varphi^*.$$

From (2.4)–(2.7) we get the optimality system (the necessary optimality conditions, Lions, 1968):

$$\begin{cases} \dfrac{\partial \varphi}{\partial t} &= F(\varphi,\lambda) + f, \quad t \in (0,T), \\ \varphi \big|_{t=0} &= u, \end{cases} \tag{2.8}$$

$$\begin{cases} \dfrac{\partial \varphi^*}{\partial t} + (F'_\varphi(\varphi,\lambda))^* \varphi^* &= C^* V_2 (C\varphi - \varphi_{obs}), \\ \varphi^* \big|_{t=T} &= 0, \end{cases} \tag{2.9}$$

$V_1(\lambda - \lambda_b) - (F'_\lambda(\varphi,\lambda))^* \varphi^* = 0.$                                                       (2.10)

We assume that the system (2.8)–(2.10) has a unique solution. The system (2.8)–(2.10) may be considered as a generalized model $\mathcal{A}(U) = 0$ with the state variable $U = (\varphi, \varphi^*, \lambda)$, and it contains information about observations. In what follows we study the problem of the sensitivity of functionals of the optimal solution to the observation data.

If the observation operator $C$ is nonlinear, i.e. $C\varphi = C(\varphi)$, then the right-hand side of the adjoint equation (2.9) contains

$(C'_\varphi)^*$ instead of $C^*$ and all the analysis presented below is similar.

## 3   Sensitivity of functionals after assimilation

In geophysical applications the observation data cannot be measured precisely, therefore, it is important to be able to estimate the impact of uncertainties in observations on the outputs of the model after assimilation.

Let us introduce a response function $G(\varphi,\lambda)$, which is supposed to be a real-valued function and can be considered as a

functional on $Y \times Y_p$. We are interested in the sensitivity of $G$ with respect to $\varphi_{obs}$, with $\varphi$ and $\lambda$ obtained from the optimality system (2.8)–(2.10). By definition, the sensitivity is defined by the gradient of $G$ with respect to $\varphi_{obs}$:

$$\frac{dG}{d\varphi_{obs}} = \frac{\partial G}{\partial \varphi} \frac{\partial \varphi}{\partial \varphi_{obs}} + \frac{\partial G}{\partial \lambda} \frac{\partial \lambda}{\partial \varphi_{obs}}. \tag{3.1}$$

If $\delta\varphi_{obs}$ is a perturbation on $\varphi_{obs}$, we get from the optimality system:

$$\begin{cases} \dfrac{\partial \delta\varphi}{\partial t} &= F'_\varphi(\varphi,\lambda)\delta\varphi + F'_\lambda(\varphi,\lambda)\delta\lambda, \\ \delta\varphi \big|_{t=0} &= 0, \end{cases} \tag{3.2}$$

$$\begin{cases} -\dfrac{\partial \delta\varphi^*}{\partial t} - (F_\varphi'(\varphi,\lambda))^* \delta\varphi^* - (F_{\varphi\varphi}''(\varphi,\lambda)\delta\varphi)^*\varphi^* &= (F_{\varphi\lambda}''(\varphi,\lambda)\delta\lambda)^*\varphi^* - C^*V_2(C\delta\varphi - \delta\varphi_{obs}), \\ \delta\varphi^*\big|_{t=T} &= 0, \end{cases}$$

(3.3)

$$V_1\delta\lambda - (F_{\lambda\varphi}''(\varphi,\lambda)\delta\varphi)^*\varphi^* - (F_{\lambda\lambda}''(\varphi,\lambda)\delta\lambda)^*\varphi^* - (F_\lambda'(\varphi,\lambda))^*\delta\varphi^* = 0,$$

(3.4)

5   and

$$\left(\dfrac{dG}{d\varphi_{obs}}, \delta\varphi_{obs}\right)_{Y_{obs}} = \left(\dfrac{\partial G}{\partial\varphi}, \delta\varphi\right)_Y + \left(\dfrac{\partial G}{\partial\lambda}, \delta\lambda\right)_{Y_p},$$

(3.5)

where $\delta\varphi$, $\delta\varphi^*$ and $\delta\lambda$ are the Gâteaux derivatives of $\varphi$, $\varphi^*$ and $\lambda$ in the direction $\delta\varphi_{obs}$ (for example, $\delta\varphi = \frac{\partial\varphi}{\partial\varphi_{obs}}\delta\varphi_{obs}$).

To compute the gradient $\nabla_{\varphi_{obs}}G(\varphi,\lambda)$, let us introduce three adjoint variables $P_1 \in Y$, $P_2 \in Y$ and $P_3 \in Y_p$. By taking the inner product of (3.2) by $P_1$, (3.3) by $P_2$ and of (3.4) by $P_3$ and adding them, we obtain:

$$\left(\dfrac{\partial\delta\varphi}{\partial t} - F_\varphi'(\varphi,\lambda)\delta\varphi - F_\lambda'(\varphi,\lambda)\delta\lambda, P_1\right)_Y + \left(-\dfrac{\partial\delta\varphi^*}{\partial t} - (F_\varphi'(\varphi,\lambda))^*\delta\varphi^* - (F_{\varphi\varphi}''(\varphi,\lambda)\delta\varphi)^*\varphi^* - \right.$$

$$\left. - (F_{\varphi\lambda}''(\varphi,\lambda)\delta\lambda)^*\varphi^* + C^*V_2(C\delta\varphi - \delta\varphi_{obs}), P_2\right)_Y + \left(V_1\delta\lambda - (F_{\lambda\varphi}''(\varphi,\lambda)\delta\varphi)^*\varphi^* - (F_{\lambda\lambda}''(\varphi,\lambda)\delta\lambda)^*\varphi^* - \right.$$

$$\left. - (F_\lambda'(\varphi,\lambda))^*\delta\varphi^*, P_3\right)_{Y_p} = 0.$$

Then, using integration by parts and adjoint operators, we get

$$\left(\delta\varphi, -\dfrac{\partial P_1}{\partial t} - (F_\varphi'(\varphi,\lambda))^*P_1 - (F_{\varphi\varphi}''(\varphi,\lambda)P_2)^*\varphi^* - (F_{\lambda\varphi}''(\varphi,\lambda)P_3)^*\varphi^* + C^*V_2CP_2\right)_Y + \left(\delta\varphi\big|_{t=T}, P_1\big|_{t=T}\right)_X +$$

$$+ \left(\delta\varphi^*, \dfrac{\partial P_2}{\partial t} - F_\varphi'(\varphi,\lambda)P_2 - F_\lambda'(\varphi,\lambda)P_3\right)_Y + \left(\delta\varphi^*\big|_{t=0}, P_2\big|_{t=0}\right)_X + \left(\delta\lambda, V_1P_3 - (F_{\varphi\lambda}''(\varphi,\lambda)P_2)^*\varphi^* - \right.$$

$$\left. - (F_{\lambda\lambda}''(\varphi,\lambda)P_3)^*\varphi^* - (F_\lambda'(\varphi,\lambda))^*P_1\right)_{Y_p} - \left(\delta\varphi_{obs}, V_2CP_2\right)_{Y_{obs}} = 0.$$

(3.6)

Here we put

$$-\dfrac{\partial P_1}{\partial t} - (F_\varphi'(\varphi,\lambda))^*P_1 - (F_{\varphi\varphi}''(\varphi,\lambda)P_2)^*\varphi^* - (F_{\lambda\varphi}''(\varphi,\lambda)P_3)^*\varphi^* + C^*V_2CP_2 = \dfrac{\partial G}{\partial\varphi},$$

and

$$V_1P_3 - (F_{\varphi\lambda}''(\varphi,\lambda)P_2)^*\varphi^* - (F_{\lambda\lambda}''(\varphi,\lambda)P_3)^*\varphi^* - (F_\lambda'(\varphi,\lambda))^*P_1 = \dfrac{\partial G}{\partial\lambda},\ P_1\big|_{t=T} = 0,$$

$$\dfrac{\partial P_2}{\partial t} - F_\varphi'(\varphi,\lambda)P_2 - F_\lambda'(\varphi,\lambda)P_3 = 0,\ P_2\big|_{t=0} = 0.$$

10   Thus, if $P_1, P_2, P_3$ are the solutions of the following system of equations

$$\begin{cases} -\dfrac{\partial P_1}{\partial t} - (F_\varphi'(\varphi,\lambda))^*P_1 - (F_{\varphi\varphi}''(\varphi,\lambda)P_2)^*\varphi^* &= (F_{\lambda\varphi}''(\varphi,\lambda)P_3)^*\varphi^* - C^*V_2CP_2 + \dfrac{\partial G}{\partial\varphi}, \\ P_1\big|_{t=T} &= 0, \end{cases}$$

(3.7)

$$\begin{cases} \dfrac{\partial P_2}{\partial t} - F_\varphi'(\varphi,\lambda)P_2 - F_\lambda'(\varphi,\lambda)P_3 &= 0, \quad t \in (0,T) \\ \left. P_2 \right|_{t=0} &= 0, \end{cases} \tag{3.8}$$

$$V_1 P_3 - (F_{\varphi\lambda}''(\varphi,\lambda)P_2)^*\varphi^* - (F_{\lambda\lambda}''(\varphi,\lambda)P_3)^*\varphi^* - (F_\lambda'(\varphi,\lambda))^*P_1 = \frac{\partial G}{\partial \lambda}, \tag{3.9}$$

then from (3.6) we get

$$\left( \frac{\partial G}{\partial \varphi}, \delta\varphi \right)_Y + \left( \frac{\partial G}{\partial \lambda}, \delta\lambda \right)_{Y_p} = \left( \delta\varphi_{obs}, V_2 C P_2 \right)_{Y_{obs}},$$

and due to (3.5) the gradient of $G$ is given by

$$\frac{dG}{d\varphi_{obs}} = V_2 C P_2. \tag{3.10}$$

We get a coupled system of two differential equations (3.7) and (3.8) of the first order with respect to time, and (3.9). To study this non-standard problem (3.7)–(3.9), we reduce it to a single operator equation involving the Hessian of the original cost function.

## 4   Operator equation via Hessian and response function gradient

Let us denote the auxiliary variable $v = P_3$ and rewrite the non-standard problem (3.7)–(3.9) in an equivalent form:

$$\begin{cases} \dfrac{\partial P_2}{\partial t} - F_\varphi'(\varphi,\lambda)P_2 &= F_\lambda'(\varphi,\lambda)v, \\ \left. P_2 \right|_{t=0} &= 0, \end{cases} \tag{4.1}$$

$$\begin{cases} -\dfrac{\partial P_1}{\partial t} - (F_\varphi'(\varphi,\lambda))^*P_1 - (F_{\varphi\varphi}''(\varphi,\lambda)P_2)^*\varphi^* &= (F_{\lambda\varphi}''(\varphi,\lambda)v)^*\varphi^* - C^*V_2 C P_2 + \dfrac{\partial G}{\partial \varphi}, \\ \left. P_1 \right|_{t=T} &= 0, \end{cases} \tag{4.2}$$

$$V_1 v - (F_{\varphi\lambda}''(\varphi,\lambda)P_2)^*\varphi^* - (F_{\lambda\lambda}''(\varphi,\lambda)v)^*\varphi^* - (F_\lambda'(\varphi,\lambda))^*P_1 = \frac{\partial G}{\partial \lambda}, \tag{4.3}$$

Here we have three unknowns: $v \in Y_p$, $P_1, P_2 \in Y$. Let us write (4.1)–(4.3) in the form of an operator eqution for $v$. We define the operator $\mathcal{H}$, which acts on $w$ belonging to $Y_p$, by the successive solution of the following problems:

$$\begin{cases} \dfrac{\partial \phi}{\partial t} - F_\varphi'(\varphi,\lambda)\phi &= F_\lambda'(\varphi,\lambda)w, \quad t \in (0,T) \\ \left. \phi \right|_{t=0} &= 0, \end{cases} \tag{4.4}$$

$$\begin{cases} -\dfrac{\partial \phi^*}{\partial t} - (F_\varphi'(\varphi,\lambda))^*\phi^* - (F_{\varphi\varphi}''(\varphi,\lambda)\phi)^*\varphi^* &= (F_{\lambda\varphi}''(\varphi,\lambda)w)^*\varphi^* - C^*V_2 C\phi, \\ \left. \phi^* \right|_{t=T} &= 0, \end{cases} \tag{4.5}$$

$$\mathcal{H}w = V_1 w - (F_{\varphi\lambda}''(\varphi,\lambda)\phi)^*\varphi^* - (F_{\lambda\lambda}''(\varphi,\lambda)w)^*\varphi^* - (F_\lambda'(\varphi,\lambda))^*\phi^*. \tag{4.6}$$

Here $\lambda, \varphi$ and $\varphi^*$ are the solutions of the optimality system (2.8)–(2.10). Then (4.1)–(4.3) is equivalent to the following equation in $Y_p$:

$$\mathcal{H}v = \mathcal{F} \tag{4.7}$$

with the right-hand side $\mathcal{F}$ defined by

$$\mathcal{F} = \frac{\partial G}{\partial \lambda} + (F_\lambda'(\varphi, \lambda))^* \tilde{\phi}^*, \tag{4.8}$$

where $\tilde{\phi}^*$ is the solution to the adjoint problem:

$$\begin{cases} -\dfrac{\partial \tilde{\phi}^*}{\partial t} - (F_\varphi'(\varphi, \lambda))^* \tilde{\phi}^* &= \dfrac{\partial G}{\partial \varphi}, \quad t \in (0, T) \\ \tilde{\phi}^*\big|_{t=T} &= 0. \end{cases} \tag{4.9}$$

It is easily seen that the operator $\mathcal{H}$ defined by (4.4)–(4.6) is the Hessian of the original functional $J$ considered on the optimal solution $\lambda$ of the problem (2.8)–(2.10): $J''(\lambda) = \mathcal{H}$. Under the assumption that $\mathcal{H}$ is positive definite, the operator equation (4.7) is correctly and everywhere solvable in $Y_p$ (Vainberg, 1964), i.e. for every $\mathcal{F}$ there exists a unique solution $v \in Y_p$ and

$$\|v\|_{Y_p} \le c\|\mathcal{F}\|_{Y_p}, \quad c = const > 0.$$

Therefore, under the assumption that $J''(\lambda)$ is positive definite on the optimal solution, the non-standard problem (3.7)–(3.9) has a unique solution $P_1, P_2 \in Y, P_3 \in Y_p$.

Based on the above consideration, we can formulate the following algorithm to compute the gradient of the response function $G$:

1) For $\dfrac{\partial G}{\partial \lambda} \in Y_p, \dfrac{\partial G}{\partial \varphi} \in Y$ solve the adjoint problem

$$\begin{cases} -\dfrac{\partial \tilde{\phi}^*}{\partial t} - (F_\varphi'(\varphi, \lambda))^* \tilde{\phi}^* &= \dfrac{\partial G}{\partial \varphi}, \\ \tilde{\phi}^*\big|_{t=T} &= 0 \end{cases} \tag{4.10}$$

and put

$$\mathcal{F} = \frac{\partial G}{\partial \lambda} + (F_\lambda'(\varphi, \lambda))^* \tilde{\phi}^*.$$

2) Find $v$ by solving

$$\mathcal{H}v = \mathcal{F}$$

with the Hessian of the original functional $J$ defined by (4.4)–(4.6).

3) Solve the direct problem

$$\begin{cases} \dfrac{\partial P_2}{\partial t} - F_\varphi'(\varphi, \lambda) P_2 &= F_\lambda'(\varphi, \lambda)v, \quad t \in (0, T) \\ P_2\big|_{t=0} &= 0. \end{cases} \tag{4.11}$$

4) Compute the gradient of the response function as

$$\frac{dG}{d\varphi_{obs}} = V_2 C P_2. \tag{4.12}$$

Formula (4.12) allows us to estimate the sensitivity of the functionals related to the optimal solution after assimilation, with respect to observation data.

**Remark 1.** In the above consideration, to show the solvability, we have assumed that the direct and adjoint tangent linear problems of the form

$$\begin{cases} \dfrac{\partial \phi}{\partial t} - F'_\varphi(\varphi,\lambda)\phi & = & f, \quad t \in (0,T) \\ \phi\big|_{t=0} & = & 0, \end{cases}$$

$$\begin{cases} -\dfrac{\partial \phi^*}{\partial t} - (F'_\varphi(\varphi,\lambda))^*\phi^* & = & g, \quad t \in (0,T) \\ \phi^*\big|_{t=T} & = & 0 \end{cases}$$

with $f, g \in Y$ have the unique solutions $\phi, \phi^* \in Y$.

**Remark 2.** The analysis presented above is based on the hypothesis that the initial state of the system under observation is known, and that it is only model parameters (boundary conditions, forcing terms, distributed coefficients, etc.) that are to be determined from the observations. Often, a more realistic situation would be one where the assimilation is intended at determining both the initial conditions of the system and, in addition, model parameters (Dee, 2005; Smith et al., 2013). The sensitivity analysis can be applied as well to such a situation. To consider joint state and parameter estimation problem, we should use the results both of this paper and of the previous one (Shutyaev et al., 2017). In this case we need to introduce an additional term related to the initial condition into the cost function (2.2) to find simultaneously $u$ and $\lambda$. The optimality system (2.8)–(2.10) will be supplemented by an additional equation related to the gradient of the cost function with respect to $u$. The Hessian in this case is a 2x2 operator-matrix, acting on the augmented vector $U = (u,\lambda)^T$, and all the derivations are made similarly, being, however, more cumbersome and lengthy.

Below we give a proof-of-concept analytic example to show how the algorithm (4.10)–(4.12) works. Then, as an application, we consider a variational data assimilation problem for a sea thermodynamics model.

## 5 Proof-of-concept analytic example

Let us consider a simple evolution problem for the ordinary differential equation

$$\begin{cases} \dfrac{d\varphi}{dt} + a\varphi = \lambda g, \quad t \in (0,T) \\ \varphi\big|_{t=0} = u, \end{cases} \tag{5.1}$$

where $u \in \mathbb{R}$; $a, \lambda \in \mathbb{R}$, $g = g(t) \geq 0$. Here, in the notations of section 2, we have $X = \mathbb{R}$, $Y = L_2(0, T)$, $F(\varphi, \lambda) = -a\varphi + \lambda g$. Let us formulate the data assimilation problem to find the parameter $\lambda$ if we have observation data for $\varphi$ at the end of the time interval $t = T$. We need to minimize the cost function

$$J(\lambda) = \inf_{v \in \mathbb{R}} J(v), \tag{5.2}$$

5    where $J(v) = \dfrac{1}{2}|\tilde{\varphi}|_{t=T} - \varphi_{obs}|^2$, and $\tilde{\varphi}$ is the solution to (5.1) with $\lambda = v$.

Thus, here we have $Y_p = \mathbb{R}$, $V_1 = 0$, $V_2 = 1$, $C\varphi = \varphi|_{t=T}$.

In this case, the optimality system (2.8)–(2.10) has the form:

$$\begin{cases} \dfrac{d\varphi}{dt} + a\varphi = \lambda g, & t \in (0, T) \\ \varphi|_{t=0} = u, \end{cases} \tag{5.3}$$

$$\begin{cases} \dfrac{d\varphi^*}{dt} - a\varphi^* = 0, & t \in (0, T) \\ \varphi^*|_{t=T} = \varphi|_{t=T} - \varphi_{obs}, \end{cases} \tag{5.4}$$

$$(g, \varphi^*) = \int_0^T g(t)\varphi^*(t) = 0. \tag{5.5}$$

It is easy to see that the problem of data assimilation (5.1)-(5.2) has a unique solution

$$\lambda = \lambda_{opt} = \frac{\varphi_{obs} - \varphi_0}{\varphi_1}, \tag{5.6}$$

15    where $\varphi_0 = u_0 e^{-aT}$, $\varphi_1 = \int_0^T e^{-a(T-t')} g(t') dt'$.

Indeed, if $\lambda$ has the form (5.6), the solution of the problem (5.1) satisfies $\varphi|_{t=T} = \varphi_{obs}$, and the functional $J$ from (5.2) attains its minimal value $J = 0$. In this case $\varphi^* = 0$, and the optimality system (5.3)-(5.5) is satisfied.

Let us consider the response function in the form

$$G(\varphi, \lambda) = \int_0^T \varphi(t) dt. \tag{5.7}$$

20    Let $a \neq 0$. After assimilation, taking into account the solution of the problem (5.1), we have

$$G(\varphi, \lambda) = \frac{u}{a}(1 - e^{-aT}) + \frac{\lambda_{opt}}{a}\left(\int_0^T g(t) dt - \varphi_1\right), \tag{5.8}$$

where $\lambda_{opt}$ is given by (5.6). Then, by differentiation of $G$ with respect to $\varphi_{obs}$ we have the gradient

$$\frac{dG}{d\varphi_{obs}} = \frac{1}{a\varphi_1}\left(\int_0^T g(t) dt - \varphi_1\right). \tag{5.9}$$

Let us now apply the algorithm (4.10)–(4.12) to compute the gradient of the function $G$. Since $\dfrac{\partial G}{\partial \varphi} = 1$, $(F'_\varphi(\varphi, \lambda))^* = -a$, then on the first step of the algorithm, we solve the problem (4.10) and get the solution

$$\tilde{\phi}^*(t) = \frac{1}{a}(1 - e^{-a(T-t)}). \tag{5.10}$$

Taking into account that $\partial G/\partial \lambda = 0$ and $(F'_\varphi(\varphi, \lambda))^* \tilde{\phi}^* = (g, \tilde{\phi}^*)$, we get $\mathcal{F} = (g, \tilde{\phi}^*)$ i.e.,

$$\mathcal{F} = \int_0^T g\tilde{\phi}^* dt = \frac{1}{a} \left( \int_0^T g(t) dt - \varphi_1 \right). \tag{5.11}$$

On the second step of the algorithm, one need to solve the equation $\mathcal{H}v = \mathcal{F}$ with the Hessian $\mathcal{H}$ defined by the formulas (4.4)–(4.6). Since all the second order derivatives of $F(\varphi, \lambda)$ equal zero, then it is easily seen that $\mathcal{H}$ in this case is the operator of multiplication by the scalar

$$\mathcal{H} = \int_0^T g(t)\psi|_{t=T} e^{-a(T-t)} dt = (\psi|_{t=T})^2, \tag{5.12}$$

where $\psi(t)$ is the solution of the problem (5.1) with $u = 0$, $\lambda = 1$.

Then, after the second step of the algorithm we get

$$v = \mathcal{H}^{-1}\mathcal{F} = (\psi|_{t=T})^{-2}\mathcal{F}. \tag{5.13}$$

On the third step of the algorithm, we need to solve the problem (4.11). Since $F'_\lambda(\varphi, \lambda) = g$, the solution of this problem has the form $P_2(t) = v\psi(t)$. Finally, using (4.12), we get the gradient of $G$ with respect to $\varphi_{obs}$:

$$\frac{dG}{d\varphi_{obs}} = P_2|_{t=T} = v\psi(T) = \frac{\psi(T)\mathcal{F}}{\psi^2(T)} = \frac{\mathcal{F}}{\psi(T)}. \tag{5.14}$$

Moreover, since $\varphi_1 = \psi(T)$, then from (5.14) and (5.11) we have

$$\frac{dG}{d\varphi_{obs}} = \frac{1}{a\varphi_1} \left( \int_0^T g(t) dt - \varphi_1 \right). \tag{5.15}$$

Thus, the gradient obtained by the algorithm (4.10)–(4.12) exactly coincides with the value of the gradient obtained in (5.9) by direct differentiation, which is the expected result.

## 6  Data assimilation problem for a sea thermodynamics model

Consider the sea thermodynamics problem in the form (Marchuk et al., 1987):

$$T_t + (\bar{U}, \mathrm{Grad})T - \mathrm{Div}(\hat{a}_T \cdot \mathrm{Grad}\, T) = f_T \ \text{ in } \ D \times (t_0, t_1),$$

$$T = T_0 \ \text{for } t = t_0 \text{ in } D,$$

$$-\nu_T \frac{\partial T}{\partial z} = Q \text{ on } \Gamma_S \times (t_0, t_1), \quad \frac{\partial T}{\partial n} = 0 \text{ on } \Gamma_{w,c} \times (t_0, t_1),$$

$$\bar{U}_n^{(-)}T + \frac{\partial T}{\partial n} = Q_T \text{ on } \Gamma_{w,op} \times (t_0, t_1),$$

$$\frac{\partial T}{\partial n} = 0 \text{ on } \Gamma_H \times (t_0, t_1), \tag{6.1}$$

where $T = T(x, y, z, t)$ is an unknown temperature function, $t \in (t_0, t_1)$, $(x, y, z) \in D = \Omega \times (0, H)$, $\Omega \subset R^2$, $H = H(x, y)$ is the function of the bottom releif, $Q = Q(x, y, t)$ is the total heat flux, $\bar{U} = (u, v, w)$, $\widehat{a}_T = \text{diag}((a_T)_{ii})$, $(a_T)_{11} = (a_T)_{22} = \mu_T$, $(a_T)_{33} = \nu_T$, $f_T = f_T(x, y, z, t)$ are given functions. The boundary of the domain $\Gamma \equiv \partial D$ is represented as a union of four disjoint parts $\Gamma_S$, $\Gamma_{w,op}$, $\Gamma_{w,c}$, $\Gamma_H$, where $\Gamma_S = \Omega$ (the unperturbed sea surface), $\Gamma_{w,op}$ is the liquid (open) part of vertical lateral boundary, $\Gamma_{w,c}$ is the solid part of the vertical lateral boundary, $\Gamma_H$ is the sea bottom, $\bar{U}_n^{(-)} = (|\bar{U}_n| - \bar{U}_n)/2$, and $\bar{U}_n$ is the normal component of $\bar{U}$. The other notations and a detailed description of the problem statement can be found in Agoshkov et al. (2008).

Problem (6.1) can be written in the form of an operator equation:

$$T_t + LT = \mathcal{F} + BQ, \quad t \in (t_0, t_1),$$
$$T = T_0, \quad t = t_0, \tag{6.2}$$

where the equality is understood in the weak sense, namely,

$$(T_t, \widehat{T}) + (LT, \widehat{T}) = \mathcal{F}(\widehat{T}) + (BQ, \widehat{T}) \ \forall \widehat{T} \in W_2^1(D), \tag{6.3}$$

in this case $L, \mathcal{F}, B$ are defined by the following relations:

$$(LT, \widehat{T}) \equiv \int_D (-T\text{Div}(\bar{U}\widehat{T}))dD + \int_{\Gamma_{w,op}} \bar{U}_n^{(+)}T\widehat{T}d\Gamma + \int_D \widehat{a}_T\text{Grad}(T) \cdot \text{Grad}(\widehat{T})dD,$$

$$\mathcal{F}(\widehat{T}) = \int_{\Gamma_{w,op}} Q_T\widehat{T}d\Gamma + \int_D f_T\widehat{T}dD, \quad (T_t, \widehat{T}) = \int_D T_t\widehat{T}dD, \quad (BQ, \widehat{T}) = \int_\Omega Q\widehat{T}\big|_{z=0}d\Omega,$$

and the functions $\widehat{a}_T$, $Q_T$, $f_T$, $Q$ are such that equality (6.3) makes sense. The properties of the operator $L$ were studied by Agoshkov et al. (2008).

Due to (6.3), the equation (6.2) is considered in $Y^* = L_2(t_0, t_1; (W_2^1(D))^*)$, and the operator $B : L_2(\Omega \times (t_0, t_1)) \to Y^*$ maps the function $Q \in L_2(\Omega \times (t_0, t_1))$ into the function $BQ \in Y^*$ such that $(BQ, \widehat{T}) = \int_\Omega Q\widehat{T}\big|_{z=0}d\Omega, \ \forall \widehat{T} \in W_2^1(D)$. Therefore, $BQ$ is a linear and bounded functional on $L_2(0, T; W_2^1(D))$.

Consider the data assimilation problem for the sea surface temperature (see Agoshkov et al., 2008). Suppose that the function $Q \in L_2(\Omega \times (t_0, t_1))$ is unknown in problem (6.1). Let also $T_{\text{obs}}(x, y, t)$ be the function on $\bar{\Omega} \equiv \Omega \cup \partial\Omega$ obtained for $t \in (t_0, t_1)$ by processing the observation data, and this function in its physical sense is an approximation to the surface temperature function on $\Omega$, i.e. to $T\big|_{z=0}$. We suppose that $T_{\text{obs}} \in L_2(\Omega \times (t_0, t_1))$, but the function $T_{\text{obs}}$ may not possess greater smoothness

and hence it cannot be used for the boundary condition on $\Gamma_S$. We admit the case when $T_{\text{obs}}$ is defined only on some subset of $\Omega \times (t_0, t_1)$ and denote the indicator (characteristic) function of this set by $m_0$. For definiteness sake, we assume that $T_{\text{obs}}$ is zero outside this subset.

Consider the data assimilation problem for the surface temperature in the following form: find $T$ and $Q$ such that

$$
\begin{cases}
T_t + LT &= \mathcal{F} + BQ \ \text{ in } D \times (t_0, t_1), \\
T &= T_0, \quad t = t_0 \\
J(Q) &= \inf_v J(v),
\end{cases}
\tag{6.4}
$$

where

$$
J(Q) = \frac{\alpha}{2} \int\limits_{t_0}^{t_1} \int\limits_{\Omega} |Q - Q^{(0)}|^2 d\Omega dt + \frac{1}{2} \int\limits_{t_0}^{t_1} \int\limits_{\Omega} m_0 |T|_{z=0} - T_{obs}|^2 d\Omega dt,
\tag{6.5}
$$

and $Q^{(0)} = Q^{(0)}(x, y, t)$ is a given function, $\alpha = const > 0$.

For $\alpha > 0$ this variational data assimilation problem has a unique solution. The existence of the optimal solution follows from the classic results of the theory of optimal control problems (Lions, 1968), because it is easy to show that the solution to problem (6.1) continuously depends on the flux $Q$ (*a priori* estimates are valid in the corresponding functional spaces), the functional $J$ is weakly lower semicontinuous, and the space of admissible controls $L_2(\Omega \times (t_0, t_1))$ is weakly compact.

For $\alpha = 0$ the problem does not always have a solution, but, as was shown by Agoshkov et al. (2008), there is unique and dense solvability, and it allows one to construct a sequence of regularized solutions minimizing the functional, which is related to a sequence of coefficients $\alpha_n$, with $\alpha_n \to 0$ when $n \to \infty$.

The optimality system determining the solution of the formulated variational data assimilation problem according to the necessary condition $\text{grad} J = 0$ has the form:

$$
T_t + LT = \mathcal{F} + BQ \quad \text{in } D \times (t_0, t_1),
$$
$$
T = T_0, \quad t = t_0,
\tag{6.6}
$$

$$
-(T^*)_t + L^* T^* = B m_0 (T_{\text{obs}} - T) \text{ in } D \times (t_0, t_1),
$$
$$
T^* = 0, \quad t = t_1,
\tag{6.7}
$$

$$
\alpha(Q - Q^{(0)}) - T^* = 0 \quad \text{on } \Omega \times (t_0, t_1),
\tag{6.8}
$$

where $L^*$ is the operator adjoint to $L$.

Here the boundary-value function $Q$ plays the role of $\lambda$ from Section 2, $\varphi = T$, the operator $F$ has the form $F(T, Q) = -LT + BQ$, and $F'_T = -L, F'_Q = B$. Since the operator $F(T, Q)$ is bilinear in this case, the Hessian $\mathcal{H}$ acting on some $\psi \in L_2(\Omega \times (t_0, t_1))$ is defined by the successive solution of the following problems:

$$
\begin{cases}
\dfrac{\partial \phi}{\partial t} + L\phi &= B\psi, \quad t \in (t_0, t_1) \\
\phi|_{t=t_0} &= 0,
\end{cases}
\tag{6.9}
$$

$$\begin{cases} -\dfrac{\partial \phi^*}{\partial t} + L^* \phi^* &= -Bm_0\phi, \quad t \in (t_0, t_1) \\ \phi^*\big|_{t=t_1} &= 0, \end{cases} \tag{6.10}$$

$$\mathcal{H}\psi = \alpha\psi - B^*\phi^*. \tag{6.11}$$

To illustrate the above-presented theory, we consider the problem of sensitivity of functionals of the optimal solution $Q$ to the observations $T_{obs}$. Let us introduce the following functional (response function):

$$G(T) = \int\limits_{t_0}^{t_1} dt \int\limits_{\Omega} k(x,y,t) T(x,y,0,t) d\Omega, \tag{6.12}$$

where $k(x,y,t)$ is a weight function related to the temperature field on the sea surface $z = 0$. For example, if we are interested in the mean temperature of a specific region of the sea $\omega$ for $z = 0$ in the interval $\bar{t} - \tau \le t \le \bar{t}$, then as $k$ we take the function

$$k(x,y,t) = \begin{cases} 1\big/(\tau \mathrm{mes}\,\omega) & \text{if } (x,y) \in \omega, \ \bar{t} - \tau \le t \le \bar{t} \\ 0 & \text{else}, \end{cases} \tag{6.13}$$

where $\mathrm{mes}\,\omega$ denotes the area of the region $\omega$. Thus, the functional (6.12) is written in the form:

$$G(T) = \frac{1}{\tau} \int\limits_{\bar{t}-\tau}^{\bar{t}} dt \left( \frac{1}{\mathrm{mes}\,\omega} \int\limits_{\omega} T(x,y,0,t) d\Omega \right). \tag{6.14}$$

Formula (6.14) represents the mean temperature averaged over the time interval $\bar{t} - \tau \le t \le \bar{t}$ for a given region $\omega$. The functionals of this type are of most interest in the theory of climate change (Marchuk, 1995; Marchuk et al., 1996).

In our notations the functional (6.12) may be written as

$$G(T) = \int\limits_{t_0}^{t_1} (Bk, T) dt = (Bk, T)_Y, \ \ Y = L_2(D \times (t_0, t_1)).$$

We are interested in the sensitivity of the functional $G(T)$, obtained for $T$ after data assimilation, with respect to the observation function $T_{obs}$.

By definition, the sensitivity is given by the gradient of $G$ with respect to $T_{obs}$:

$$\frac{dG}{dT_{obs}} = \frac{\partial G}{\partial T} \frac{\partial T}{\partial T_{obs}}. \tag{6.15}$$

Since $\frac{\partial G}{\partial T} = Bk$, then according to the theory presented in Section 4, to compute the gradient (6.15) we need to perform the following steps:

1) For $k$ defined by (6.13) solve the adjoint problem

$$\begin{cases} -\dfrac{\partial \tilde{\phi}^*}{\partial t} + L^* \tilde{\phi}^* &= Bk, \quad t \in (t_0, t_1) \\ \tilde{\phi}^*\big|_{t=t_1} &= 0 \end{cases} \tag{6.16}$$

and put $\Phi = B^* \tilde{\phi}^*$.

2) Find $\chi$ by solving $\mathcal{H}\chi = \Phi$ with the Hessian defined by (6.9)–(6.11).

3) Solve the direct problem

$$
\begin{cases}
\dfrac{\partial P_2}{\partial t} + L P_2 &= B\chi, \quad t \in (t_0, t_1) \\
P_2\big|_{t=t_0} &= 0.
\end{cases}
\tag{6.17}
$$

4) Compute the gradient of the response function as

$$
\frac{dG}{dT_{obs}} = m_0 P_2 \big|_{z=0}.
\tag{6.18}
$$

Formula (6.18) allows us to estimate the sensitivity of the functionals related to the mean temperature after data assimilation, with respect to the observations on the sea surface.

## 7 Numerical example for the Baltic Sea dynamics model

The numerical experiments have been performed using the three-dimensional numerical model of the Baltic Sea hydrother-modynamics developed at the INM RAS on the base of the splitting method (Zalesny et al., 2017) and supplied with the assimilation procedure (Agoshkov et al., 2008) for the surface temperature $T_{obs}$ with the aim to reconstruct the heat fluxes $Q$.

The object of simulation is the Baltic Sea water area. The parameters of the considered domain and its geographic coordinates can be described in the following way: $\sigma$-grid is $336 \times 394 \times 25$ (the latitude, longitude, and depth, respectively). The first point of the "grid C" (Zalesny et al., 2017) has the coordinates $9.406°$ E and $53.64°$ N. The mesh sizes in $x$ and $y$ are constant and equal to 0.0625 and 0.03125 degrees. The time step is $\Delta t = 5$ minutes. The initial condition for the whole model, including $T_0$, was chosen in the following way: the model was start running with zero initial conditions and ran with atmospheric forcing obtained from reanalysis about 20 years, and after that the result of calculation was taken as an initial condition for further running of the model. The assimilation procedure worked only during some time windows. To start the assimilation procedure for the heat flux estimation, the initial condition was taken as a model forecast for the previous time interval.

The Baltic Sea daily-averaged nighttime surface temperature data were used for $T_{obs}$. These are the data of the Danish Meteorological Institute based on measurements of radiometers (AVHRR, AATSR and AMSRE) and spectroradiometers (SEVIRI and MODIS) (Karagali, 2012). Data interpolation algorithms were used (Zakharova et al., 2013) to convert observations on computational grid of the numerical model of the Baltic Sea thermodynamics. On each time step the heat flux was determined at each surface point, therefore, the number of scalar parameters to be determined were equal to the number of scalar observations.

The mean climatic flux obtained from the NCEP (National Center for Environmental Prediction) reanalysis was taken for $Q^{(0)}$. We need to mention that $Q^{(0)}$ has a physical meaning here, it is not only an initial guess, but a parameter calculated from atmospheric data and taken in the model for temperature boundary condition on the sea surface when the model runs without assimilation procedure.

Using the hydrothermodynamics model mentioned above, which is supplied with the assimilation procedure for the surface temperature $T_{obs}$, we have performed calculations for the Baltic Sea area where the assimilation algorithm worked only at certain time moments $t_0$; in this case $t_1 = t_0 + \Delta t$. The aim of the experiment was the numerical study of the sensitivity of functionals of the optimal solution $Q$ to observation errors in the interval $(t_0, t_1)$.

Implementing the assimilation procedure, we considered a system of form (6.6)–(6.8), where (6.6)–(6.7) mean the finite-dimensional analogues of the corresponding problems (Agoshkov et al., 2008). For the statement of a data assimilation problem we introduce the cost function (6.5) with a regularization parameter $\alpha$, which weights the squared difference $|Q - Q^{(0)}|^2$. Since in all numerical experiments $\alpha$ was chosen very small, the impact of the first term in the functional was also small, and therefore $Q$ was different from $Q^{(0)}$.

We use here the SI units, namely, K (kelvin) is used for temperature, $\mathrm{ms}^{-1}$ for velocities, $\mathrm{mKs}^{-1}$ for the heat flux $Q$. The parameter $\alpha$ is defined as $\mathrm{s}^2\mathrm{m}^{-2}$ to give the both terms in (6.5) the same dimension. It is easily seen that in this case, the units of the gradient $\frac{dG}{dT_{obs}}$ from (6.18) are defined as $\mathrm{m}^{-2}\mathrm{s}^{-1}$.

Let us present some results of numerical experiments.

The calculation results for $t_0 = 50$ hours (600 time steps for the model) are presented in Fig.1 showing the gradient of the functional $G(T)$ defined by (6.14) and related to the mean temperature after data assimilation, with respect to the observations on the sea surface, according to (6.16)–(6.18). Here $\omega = \Omega$, $\tau = \Delta t$, $\bar{t} = t_1$, $\alpha = 10^{-5}\mathrm{s}^2\mathrm{m}^{-2}$.

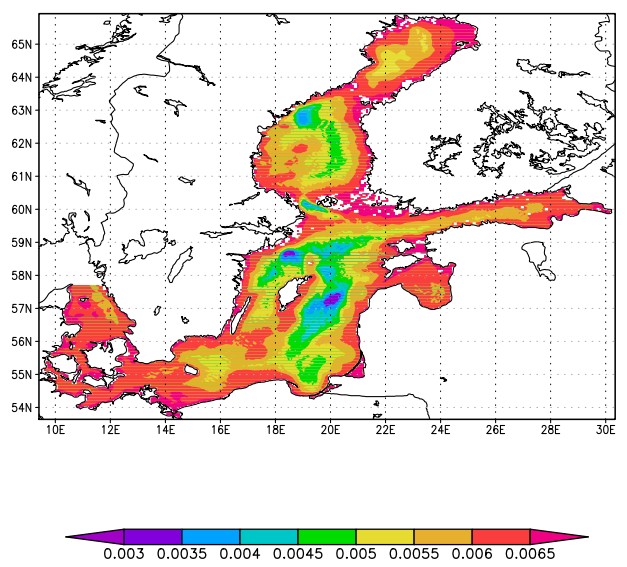

**Figure 1.** The gradient of the functional $G(T)$ $[\mathrm{m}^{-2}\mathrm{s}^{-1}]$

We can see the sub-areas (in red) in which the functional $G(T)$ is most sensitive to errors in the observations during assimilation. The largest values of the gradient of $G(T)$ correspond to the points $x, y$ lying near the regions with a small depth (cf. sea topography, Fig.2). One explanation of this phenomenon may be the fact that in the areas with depths of about $50m$, rapid

convection occurs in the upper mixed layer. With the assimilation of the surface temperature, information is transmitted faster to shallower depths, which in turn contributes to a higher sensitivity to data in these places, in contrast to deeper regions.

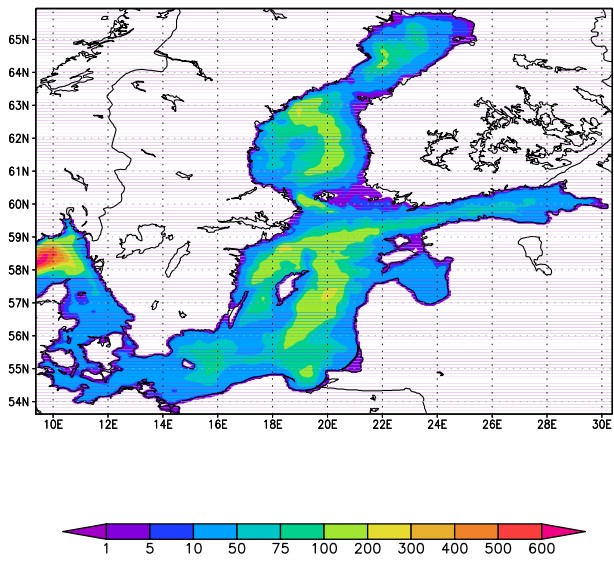

**Figure 2.** Baltic Sea topography [m]

**Remark 3.** We use the discretize-then-optimize approach, and for numerical experiments all the presented equations are understood in a discrete form, as finite-dimensional analogues of the corresponding problems, obtained after approximation. This allows us to consider model equations as a perfect model, with no approximation errors. Therefore, the accuracy of the sensitivity estimates given by the algorithm (6.16)– (6.18) are determined by the accuracy of solving the Hessian equation $\mathcal{H}\chi = \Phi$ (step 2 of the Algorithm). Due to (6.9)– (6.11), this equation is equivalent to a linear data assimilation problem, and an approximate solution to the minimization problem is obtained by an iterative procedure.

The above studies allow us to solve the problem of the definition of sea sub-areas in which the functional of the optimal solution is most sensitive to errors in the observations during variational data assimilation, when the error values are not apriori known.

## 8   Conclusions

In this paper we have considered numerical algorithms to study the sensitivity of functionals of the optimal solution of variational data assimilation problem aimed at the reconstruction of unknown parameters of the model. The optimal solution obtained as a result of assimilation depends on the observations that may contain uncertainties. Computing the gradient of the functionals with respect to observations reduces to the solution of a non-standard problem which is a coupled system in-

volving direct and adjoint equations with mutually dependent variables. Solvability of the non-standard problem is related to the properties of the Hessian of the original cost function. An algorithm developed to compute the gradient of the response function is based on the second-order adjoint techniques. Numerical example for variational data assimilation problem related to sea surface temperature for the Baltic Sea thermodynamics model demonstrates the result of the gradient computation of the response function associated with the mean surface temperature. The presented algorithm may be used to determine the sea sub-areas in which the functionals of the optimal solution are most sensitive to errors in the observations during variational data assimilation.

*Competing interests.* The authors declare that they have no conflict of interest.

*Acknowledgements.* The authors are greatly thankful to Olivier Talagrand and the reviewers for providing helpful comments that resulted in substantial improvements of the paper. This work was carried out within Russian Science Foundation project 17-77-30001 (studies in Sections 1-4), the AIRSEA Project (INRIA Grenoble Rhône-Alpes), and the project 18-01-00267 of the Russian Foundation for the Basic Research.

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
