# Peer review of "Sensitivity analysis with respect to observations in variational data assimilation for parameter estimation"

_Nonlinear Processes in Geophysics, 2018_

## Referee Comment (RC1) · Anonymous Referee #1 · 8 Mar 2018

The paper presents a study of sensitivity analysis in variational assimilation, based of the adjoint of the optimality system associated with the assimilation process (second order adjoint). It largely repeats the contents of a previous paper by Shutyaev *et al*. (2017), which has almost the same title. There are two differences in the new paper : sensitivity with respect to observations is now determined for forcing parameters (instead of initial conditions), and a numerical example, built on a model of the dynamics of the Baltic sea, is presented.

The authors write that this new paper is a generalization of the previous one. That is a bit of an exaggeration. Presenting a variant of the sensitivity analysis and a simple numerical example does not really constitute a generalization.

I suggest, before the paper can be accepted, that the authors show more clearly in what it is original, and that they also describe and discuss in more detail the numerical example that they present. It would also be desirable to extend the scope of the paper, as said just below.

1. The analysis presented in the paper is entirely based on the hypothesis that the initial state of the system under observation is known, and that it is only 'parameters' (heat fluxes in the numerical application) that are to be determined from the observations. From a physical point of view, that is highly unrealistic. I presume the authors intended their paper at being only a theoretical and numerical presentation of sensitivity of assimilation-determined parameters to observations. A much more realistic situation would be one where the assimilation is intended at determining both the initial conditions of the system and, in addition, boundary conditions and/or forcing terms. The sensitivity analysis would apply as well to such a situation. I would like to see that point discussed in some detail. In particular, could it be possible to perform an assimilation and a sensitivity analysis in those more general conditions ? I reserve my opinion on acceptance or rejection of the paper on additional information on that aspect.

And I mention that it is not said which initial conditions $T_0$ was chosen for the numerical experiment described in Section 5. That should be mentioned in case that experiment is discussed in a future version of the paper.

2. The second order approach to sensitivity analysis in variational assimilation is described in detail (and in very clear terms) in Sections 2 and 3. But, as shown in particular in the references the authors give (such as Le Dimet *et al*., 1997, in which the same approach is described), it is classical. That should be made perfectly clear, and it must be said that Sections 2 and 3 are only reminders, with possibly minor changes in detail, of already published material.

3. Concerning the numerical application, too little is said about it. Whatever you will put in a future version, a number of things have to be specified explicitly. For instance

- what was the numerical dimension of the problem (how many scalar parameters to be determined from how many scalar observations ) ?

- and say more about the results. For instance, in the present case, was the minimizing $Q$ significantly different from the guess $Q^{(0)}$ ?

- and try to interpret the results physically. In the present case, can it be explained why the sensitivity to the observations is larger in shallow areas (Fig. 1) ?

Additional remarks.

3. As said above, the title of the present paper is almost the same as the one of the paper Shutyaev *et al*. (2017). There should be more difference. I do not make suggestion at this point, since an appropriate title may depend on the content of the final paper.

4. Eqs 5.1. Is the velocity $U$ assumed to be known from the start ? What is $U_n^{(-)}$ ? And it is the same on both sides of the third equation (is there not a $U_n^{(+)}$)? And what is $d_T$ ?

5. P. 9, ll. 15-16. I understand you are referring to a sequence of coefficients $\alpha_n$, with $\alpha_n \rightarrow 0$ when $n \rightarrow \infty$.

6. P. 3, l. 10. The proper spelling is *Fréchet* (with an acute accent on the first *e*)

7. P. 2, ll. 19-20. There is a slight inconsistency of notation there. If $\varphi$ belongs to space $X$, then the operator $F$ must be defined on $X \times Y_p$. Is the derivative $\partial\varphi/\partial t$ supposed to belong to a different space ($Y$) than $\varphi$ ? And it might be useful for some readers to state explicitly that the scalar product on $Y$ involves a time integral.

P. 6, l. 8. … *the operator $\mathcal{H}$, which acts on w belonging to $Y_p$, …*

P. 8, Eq. 5.2. Say what $B$ is (the identity operator here, I presume ?).

**References**

Le Dimet, F.-X., Ngodock, H. E., Luong, B., and Verron, J.: Sensitivity analysis in variational data assimilation, *J. Meteorol. Soc. Japan*, **75** (1B), 245-255, 1997.

Shutyaev, V., Le Dimet, F.-X, and Shubina E.: Sensitivity with respect to observations in variational data assimilation, *Russ. J. Numer. Anal. Math. Modelling*, **32** (1), 61-71, 2017

---

## Referee Comment (RC2) · Anonymous Referee #2 · 2 Apr 2018

General Comments:

The manuscript presents a theoretical framework to evaluate the observation sensitivity in a variational data assimilation (VDA) system aimed at estimating model parameters. The approach relies on adjoint-modeling to derive the continuous sensitivity equations from the first order optimality system in (VDA), as put forward in a general context by Le Dimet et al. (1997). The mathematical procedure is standard, albeit lengthy, and the authors note that the equations presented here are an extension of their previous work. The theoretical results are of interest for practical applications where first- and second-order adjoint models have been developed. Some aspects need to be further clarified and the article will benefit from insertion of numerical experiments that provide an easily reproducible proof-of-concept. Further details are also needed to explain the

significance of the numerical results.

Nevertheless, in my opinion, the theoretical equations derived by authors provide a valuable, nontrivial contribution to advance the current status of science of sensitivity analysis in VDA and the manuscript may be considered for publication after revision.

Specific comments:

1. It is not clear whether the current formulation of the data assimilation system may not be simply incorporated into the previous case of initial condition estimation through a state augmentation procedure for joint state and parameter estimation, see for example, Dee (2005), Smith et al. (2013). As such, the authors should clearly state the need for the re-derivation of several equations presented here and whether the current context may not by reduced to a previously developed theory through an appropriate change in notation.

2. The significance of the numerical results is only briefly discussed and it appears that the sole purpose of the experiments is to illustrate the practical ability to evaluate the observation sensitivity in a non-trivial application. Little can be learned from these results and, in particular, important practical issues need further clarification. For example, the observation sensitivity calculations are derived from the first order optimality system however, in practice, only an approximate solution to the minimization problem is obtained through an iterative procedure. As such, solving the continuous sensitivity equations may result in inconsistencies between the optimization process and the observation sensitivity calculations. It is not clear what approach has been adopted here: discretize-then-optimize or optimize-then-discretize? Some practical issues regarding the accuracy of the sensitivity estimates should be discussed in the manuscript.

3. In my opinion, the manuscript will benefit from the insertion of a proof-of-concept with a simple model and numerical results using an easily reproducible assimilation setup where several practical aspects can be investigated and illustrated.

**References:**

Smith, P. J., Thornhill, G. D., Dance, S. L., Lawless, A. S., Mason, D. C. and Nichols, N. K. (2013), Data assimilation for state and parameter estimation: application to morphodynamic modelling. Q.J.R. Meteorol. Soc., 139: 314-327. doi:10.1002/qj.1944

Dee, D. P. (2005), Bias and data assimilation. Q.J.R. Meteorol. Soc., 131: 3323-3343. doi:10.1256/qj.05.137
* * *

---

## Author Response (AR1)

**RESPONSE TO REVIEWER 1**

The authors would like to thank the Reviewer for the time to review our paper and to provide valuable comments and suggestions. We view the criticism positively, which we address in the revised version of our manuscript. Here we would like to list our responses to each item raised by the Reviewer:

*1. The analysis presented in the paper is entirely based on the hypothesis that the initial state of the system under observation is known, and that it is only 'parameters' (heat fluxes in the numerical application) that are to be determined from the observations. From a physical point of view, that is highly unrealistic. I presume the authors intended their paper at being only a theoretical and numerical presentation of sensitivity of assimilation-determined parameters to observations. A much more realistic situation would be one where the assimilation is intended at determining both the initial conditions of the system and, in addition, boundary conditions and/or forcing terms. The sensitivity analysis would apply as well to such a situation. I would like to see that point discussed in some detail. In particular, could it be possible to perform an assimilation and a sensitivity analysis in those more general conditions ? I reserve my opinion on acceptance or rejection of the paper on additional information on that aspect.*

*And I mention that it is not said which initial conditions $T_0$ was chosen for the numerical experiment described in Section 5. That should be mentioned in case that experiment is discussed in a future version of the paper.*

We think that the parameter estimation problem is important itself. A precise determination of the initial condition is very important in view of forecasting, however the use of variational data assimilation is not limited to operational forecasting. In many domains (e.g. hydrology) the uncertainty in the parameters is more crucial that the uncertainty in the initial condition (White et al., 2003). In some problems the quantity of interest can be represented directly by the estimated controls. For example, in Agoshkov et al. (2015) the sea surface heat flux is estimated in order to understand its spatial and temporal variability:

Agoshkov V.I., Parmuzin E.I., Zalesny V.B., et al. Variational assimilation of observation data in the mathematical model of the Baltic Sea dynamics // Russ. J. Numer. Anal. Math. Modelling, 2015, v.30, no.4, pp. 203-212.

The problems of parameter estimation are common inverse problems considered in geophysics and in engineering applications:

O.M. Alifanov, E.A. Artyukhin, S.V. Rumyantsev. Extreme Methods for Solving Ill-posed Problems with Applications to Inverse Heat Transfer Problems, Begel House Publishers, 1996.
N-Z. Sun. Inverse Problems in Groundwater Modeling, Kluwer, Dordrecht, 1994.
Y. Zhu, I.M. Navon. Impact of parameter estimation on the performance of the FSU global spectral model using its full-physics adjoint. Monthly Weather Rev. 127 (1999) 1497–1517.
L.W. White, B. Vieux, D. Armand, et al. Estimation of optimal parameters for a surface hydrology model. Advances in Water Resources. 2003, V.26, No.3, pp.337-348.

R.B. Storch, L.C.G. Pimentel, and H.R.B. Orlande. Identification of atmospheric boundary layer parameters by inverse problem, Atmospheric Environment, 2007, v.41 (7), 1417-1425. doi: 10.1016/j.atmosenv.2006.10.014.

Last years an interest is arising to the parameter estimation using 4D-Var:

M. Bocquet. Parameter-field estimation for atmospheric dispersion: application to the Chernobyl accident using 4D-Var. Q. J. R. Meteorol. Soc. 2012, v.138, pp.664-681.
S. Schirber, D. Klocke, R. Pincus, J. Quaas, and J.L. Anderson. Parameter estimation using data assimilation in an atmospheric general circulation model: From a perfect toward the real world. Journal of advances in modelling Earth systems, 2013, V.5, 58–70.
C.L. Defforge, M.Bocquet, R. Bresson, P. Armand, and B. Carissimo. Data assimilation at local scale to improve CFD simulations of dispersion around industrial sites and in urban neighbourhoods. 18th International Conference on Harmonisation within Atmospheric Dispersion Modelling for Regulatory Purposes (Harmo18), 9-12 October 2017, Bologna, Italy.
W. Yuepeng, Ch. Yue, I.M. Navon, G. Yuanhong. Parameter identification techniques applied to an environmental pollution model. Journal of industrial and management optimization, 2018, 14(2): 817-831. doi: 10.3934/jimo.
Agoshkov V.I. Statement and study of some inverse problems in modelling of hydrophysical fields for water areas with `liquid' boundaries // RJNAMM, 2017. V. 32, No. 2. P. 73-90.
Agoshkov V.I., Sheloput T.O. The study and numerical solution of some inverse problems in simulation of hydrophysical fields in water areas with 'liquid' boundaries // RJNAMM, 2017. V.32, No. 3. P. 147-164.

We consider a dynamic formulation of variational data assimilation problem in a continuous form. The presented sensitivity analysis formulas do not follow from our previous results for the initial condition problem (Shutyaev et al., 2017). Of course, the initial condition function may be considered as a parameter, however, in our dynamic formulation we have 2 equations for the model: one equation for describing an evolution of the model operator (involving parameters such as right-hand sides, coefficients, boundary conditions etc.), and another equation is considered as an initial condition.

We also can consider joint state and parameter estimation problem, but it will be some generalization of this paper and the previous one (Shutyaev et al., 2017). In this case we need to introduce an additional term related to the initial condition into the cost function (2.2) to find simultaneously $u$ and $\lambda$. The optimality system (2.8)-(2.10) will be supplemented by an additional equation related to the gradient of the cost function with respect to $u$. In this case, the Hessian is a 2x2 operator-matrix, and all the derivations are more complicated, cumbersome and lengthy. Of course, we can do this, but we decided to present here only parameter estimation case, because this is the case we deal with at our Institute for numerical experiments to find the heat fluxes for the Baltic Sea thermodynamic model. In our experiments, the initial condition is supposed to be known and taken from the run of the model on the previous time step.

*2. The second order approach to sensitivity analysis in variational assimilation is described in detail (and in very clear terms) in Sections 2 and 3. But, as shown in particular in the references*

*the authors give (such as Le Dimet et al., 1997, in which the same approach is described), it is classical. That should be made perfectly clear, and it must be said that Sections 2 and 3 are only reminders, with possibly minor changes in detail, of already published material.*

Of course, we agree that the second-order approach to sensitivity analysis in variational assimilation is classical. The idea was first described by Le Dimet et al., 1997, where sensitivity with respect to model parameters was determined for the initial condition data assimilation problem. Here, we consider the same idea, but study the sensitivity with respect to observations for parameter estimation problem. The derivation of the sensitivity formulas for this very case is the novelty of our paper. We follow the logics (and even repeat some sentences) of our previous paper by Shutyaev et al. (2017), but all the formulas presented in such a form has never been published before.

*3. Concerning the numerical application, too little is said about it. Whatever you will put in a future version, a number of things have to be specified explicitly. For instance*
*- what was the numerical dimension of the problem (how many scalar parameters to be determined from how many scalar observations ) ?*
*- and say more about the results. For instance, in the present case, was the minimizing Q significantly different from the guess $Q^{(0)}$?*
*- and try to interpret the results physically. In the present case, can it be explained why the sensitivity to the observations is larger in shallow areas (Fig. 1) ?*

In the revised version of the paper we give more details concerning the numerical experiments. The three-dimensional numerical model of the Baltic Sea hydrothermodynamics developed at the Institute of Numerical Mathematics (Russian Academy of Sciences) was used in the experiments. The initial condition for the whole model, including $T_0$, was chosen in the following way: the model was start running with zero initial condition and ran with atmospheric forcing obtained from reanalysis about 20 years, and after that the result of calculation was taken as an initial condition for further running of the model. The assimilation procedure worked only during some time windows. To start the assimilation procedure for the heat flux estimation, the initial condition was taken as a model forecast for the previous time interval.

The mask water-land had the dimension 336×394 points at the surface and the number of water points where the calculations took place was 28984. Thus, the mesh size of the experiment area was $0.0625^\circ \times 0.03125^\circ$. The observation data were given on the mesh $0.03^\circ$ and were interpolated on the model grid. On each time step the heat flux was determined at each surface point, therefore, the number of scalar parameters to be determined were equal to the number of scalar observations.

We need to mention that $Q^{(0)}$ has a physical meaning here, it is not only an initial guess, but a parameter calculated from atmospheric data and taken in the model for temperature boundary condition on the sea surface when the model runs without assimilation procedure. For the statement of a data assimilation problem we introduce the cost function (5.5) with a regularization parameter α, which appears near the term involving Q and $Q^{(0)}$. Since in all

numerical experiments α was chosen very small, for instance $\alpha=10^{-5}$, the impact of the first term in the functional was also small, and therefore Q was different from $Q^{(0)}$.

The numerical experiments revealed the fact that sensitivity to the observations is larger in shallow areas (Fig. 1). One explanation of this phenomenon may be the fact that in the areas with depths of about 50 m, rapid convection occurs in the upper mixed layer. With the assimilation of the surface temperature, information is transmitted faster to shallower depths, which in turn contributes to a higher sensitivity to data in these places, in contrast to deeper regions.

*Additional remarks.*
*3. As said above, the title of the present paper is almost the same as the one of the paper Shutyaev et al. (2017). There should be more difference.*

In the revised version of the paper we correct the title as follows: "Sensitivity analysis with respect to observations in variational data assimilation for parameter estimation".

*4. Eqs 5.1. Is the velocity U assumed to be known from the start ? What is $U_n^{(-)}$ ? And it is the same on both sides of the third equation (is there not a $U_n^{(+)}$ )? And what is $d_T$ ?*

Yes, the velocity $U$ is supposed to be known, and it is stated on page 11. The definition of $U_n^{(-)}$ is given in the revised version. The parameter $d_T$ arose from another control problem, and it was taken to be zero in our numerical experiments, so we eliminate it from the statement in the revised version.

*5. P. 9, ll. 15-16. I understand you are referring to a sequence of coefficients $\alpha_n$, with $\alpha_n \to 0$ when $n \to \infty$.*

Yes, we put this explanation in the text.

*6. P. 3, l. 10. The proper spelling is Fréchet (with an acute accent on the first e).*

Corrected.

*7. P. 2, ll. 19-20. There is a slight inconsistency of notation there. If $\phi$ belongs to space X, then the operator F must be defined on X x Yp. Is the derivative $\partial\phi/\partial t$ supposed to belong to a different space (Y) than $\phi$ ? And it might be useful for some readers to state explicitly that the scalar product on Y involves a time integral.*

Corrected in the revised version.

*P. 6, l. 8. ... the operator H, which acts on w belonging to $Y_p$, ...*

Corrected in the text.

*P. 8, Eq. 5.2. Say what B is (the identity operator here, I presume ?).*

We give the explanations concerning *B* on page 11.

Most of the explanations presented here are introduced in the text of the revised version, and they are marked red there.

We are greatly thankful to the Reviewer for very useful critical remarks and comments which helped us to improve the paper and made us overlook our approach from a more relevant perspective.

**RESPONSE TO REVIEWER 2**

The authors would like to thank the Referee for reviewing our manuscript and for providing the authors with the constructive remarks and recommendations, which we have found to be very enlightening. We address the comments in the revised version of our manuscript. Here is a list of our responses to the Referee comments:

*1. It is not clear whether the current formulation of the data assimilation system may not be simply incorporated into the previous case of initial condition estimation through a state augmentation procedure for joint state and parameter estimation, see for example, Dee (2005), Smith et al. (2013). As such, the authors should clearly state the need for the re-derivation of several equations presented here and whether the current context may not by reduced to a previously developed theory through an appropriate change in notation.*

The main difference between Dee (2005), Smith et al. (2013) and our paper is that we consider a dynamic formulation of variational data assimilation problem in a continuous form. Therefore, the presented equations can not be derived directly from Dee (2005), Smith et al. (2013).
The presented sensitivity analysis formulas do not follow from our previous results for the initial condition problem (Shutyaev et al., 2017). Of course, the initial condition function may be considered as a parameter, however, in our dynamic formulation we have 2 equations for the model: one equation for describing an evolution of the model operator (involving parameters such as right-hand sides, coefficients, boundary conditions etc.), and another equation is considered as an initial condition.
We also can consider joint state and parameter estimation problem, and it will be some generalization of this paper and the previous one (Shutyaev et al., 2017). In this case we need to introduce an additional term related to the initial condition into the cost function (2.2) to find simultaneously *u* and λ. The optimality system (2.8)-(2.10) will be supplemented by an additional equation related to the gradient of the cost function with respect to *u*. In this case, the Hessian is a 2x2 operator-matrix, and all the derivations are more complicated, cumbersome and lengthy. Of course, we can do this, but we decided to present here only parameter estimation case, because this is the case we deal with at our Institute for numerical experiments to find the

heat fluxes for the Baltic Sea thermodynamic model. In our experiments, the initial condition is supposed to be known and taken from the run of the model on the previous time step.

*2. The significance of the numerical results is only briefly discussed and it appears that the sole purpose of the experiments is to illustrate the practical ability to evaluate the observation sensitivity in a non-trivial application. Little can be learned from these results and, in particular, important practical issues need further clarification. For example, the observation sensitivity calculations are derived from the first order optimality system however, in practice, only an approximate solution to the minimization problem is obtained through an iterative procedure. As such, solving the continuous sensitivity equations may result in inconsistencies between the optimization process and the observation sensitivity calculations. It is not clear what approach has been adopted here: discretize-then-optimize or optimize-then-discretize? Some practical issues regarding the accuracy of the sensitivity estimates should be discussed in the manuscript.*

In the revised version of the paper we give more details concerning the numerical experiments. We use the discretize-then-optimize approach, and for numerical experiments all the presented equations are understood in a discrete form, as finite-dimensional analogues of the corresponding problems, obtained after approximation. This allows us to consider model equations as a perfect model, with no approximation errors. Therefore, the accuracy of the sensitivity estimates given by the algorithm (5.16)-(5.18) are determined by the accuracy of solving the Hessian equation $\mathcal{H}\chi = \Phi$ (step 2 of the Algorithm). Due to (5.9)-(5.11), this equation is equivalent to a linear data assimilation problem, and an approximate solution to the minimization problem is obtained by an iterative procedure.

*3. In my opinion, the manuscript will benefit from the insertion of a proof-of-concept with a simple model and numerical results using an easily reproducible assimilation setup where several practical aspects can be investigated and illustrated.*

In the revised version of the paper we include a proof-of-concept analytic example with a simple model to demonstrate how the sensitivity analysis algorithm (5.16)-(5.18) works. Numerical analysis of the algorithm is given for a non-trivial application for the Baltic Sea dynamics model.

Most of the explanations presented here are introduced in the text of the revised version, and they are marked red there.

We are greatly thankful to the Referee for general appreciation of our work and for very useful remarks and comments which helped us to improve the paper.

Sincerely,

Victor Shutyaev, Francois-Xavier Le Dimet, and Eugene Parmuzin

---

## Referee Report (RR1)

The paper has been significantly improved. I nevertheless consider a few improvements are still necessary before it can be accepted for publication. I put my suggestions below. The first one bears on a point which has some importance, and which escaped my attention in my first review.

1. P. 15, l. 3, ... $t_1 = t_0 + \Delta t$. Does it mean the assimilation was performed over only one timestep $\Delta t = 5$ minutes of the model ? If so, that reduces somewhat the interest of the experiment. It means that there is no propagation of information between grid-points, and that the adjustment of heat fluxes to observed temperatures is purely local. In addition, no significant convection can occur over the assimilation window, thus rendering inappropriate the explanation given p. 15, ll. 16-18 for the larger sensitivities seen in shallow areas on Fig. 1. Clarification of these points is desirable.

2. Since numerical values are given without units (and without elements for comparison) in the numerical experiments (Figure 1, parameter $\alpha$ on p. 15, l. 8), they are almost meaningless (and would not allow comparison with other experiments). It would be desirable to say more. In particular were SI units used in the numerical experiments ?

3. P. 9, l. 5. Since $\lambda$ is defined in Eq. (5.2) as the minimizer of the function $J$, $J(v)$ would be more appropriate here (check for possible similar corrections elsewhere).

4. P. 4, l. 12, ... *it contains all the available information*. That is vague. Can you be more precise ?

5. P. 15, l. 7, *a regularization parameter $\alpha$, which appears near the term involving $Q$ and $Q^{(0)}$* → ... *parameter $\alpha$, which weights the squared difference* $|Q - Q^{(0)}|^2$.

6. P. 14, l. 17, ... *with zero initial condition* ... I understand this means that the initial velocity was zero. But what about temperature ? What is uniform ?

---

## Editor Decision (ED1)

Dear Victor,

As you must have seen, two referees have submitted their reports on the revised version of your paper. The referees are the same as those of the first version. In particular, I am again referee 1.

Referee 2 recommends acceptance of the paper as it stands. I have on my side a few suggestions for improvements. I think in particular that some clarification is desirable concerning the length of the assimilation window in your numerical experiment, and on the conclusions that can correspondingly be drawn.

I thank you for having submitted your paper to the NPG Special Issue in tribute to Anna Trevisan, and I look forward to receiving the final version,

Olivier Talagrand
Editor, NPG

---

## Author Response (AR2)

Dear Olivier,

Thank you very much for your valuable comments and suggestions. We take them into account in the revised version. Below are our responses to each item.

*1. P. 15, l. 3, ... t1 = t 0 + Δt. Does it mean the assimilation was performed over only one timestep Δt = 5 minutes of the model ? If so, that reduces somewhat the interest of the experiment. It means that there is no propagation of information between grid-points, and that the adjustment of heat fluxes to observed temperatures is purely local. In addition, no significant convection can occur over the assimilation window, thus rendering inappropriate the explanation given p. 15, ll. 16-18 for the larger sensitivities seen in shallow areas on Fig. 1. Clarification of these points is desirable.*

We performed the assimilation for different time intervals, and then t1=t0 + M Δt, where M is the number of subintervals we use for approximation. The number M could be sufficiently large, such that MΔt is one day, or several months. However, in Fig.1, we present the results of calculations just for M = 1, to demonstrate the work of the algorithm to construct the gradient G' for one time subinterval. Due to the splitting method, for each subinterval we assimilate $T_{obs}$ to reconstruct the heat flux Q (on the sea surface) which depends on x,y,t. It is not an initialization problem, but the parameter estimation problem, and in this case the model itself plays a role of an interpolant and propagates the information between grid points in x,y,z. Therefore, the adjustment of heat fluxes to the observed temperature is done in each grid point x,y, and the assimilation takes into account interactions in z-direction, because we use for assimilation an iterative process, and all the model parameters are consistent with the full INM RAS model (where temperature, velocities, salinity, sea level etc. are calculated at each time step). Therefore, even for a short time interval, the convection term plays a role in assimilation, and effects the result in z-direction (depth). The splitting procedure for assimilation and the iterative process used are described in detail in our former papers [1]-[2].

*2. Since numerical values are given without units (and without elements for comparison) in the numerical experiments (Figure 1, parameter α on p. 15, l. 8), they are almost meaningless (and would not allow comparison with other experiments). It would be desirable to say more. In particular were SI units used in the numerical experiments ?*

We use here the SI units, namely, K (kelvin) is used for temperature, $ms^{-1}$ for velocities, $mKs^{-1}$ for the heat flux Q. The parameter α is defined as $s^2m^{-2}$ to give the both terms in (6.5) the same dimension. It is easily seen that in this case, the units of the gradient G' from (6.18) are defined as $m^{-2}s^{-1}$. We introduce these details in the revised version.

*3. P. 9, l. 5. Since λ is defined in Eq. (5.2) as the minimizer of the function J, J(v) would be more appropriate here (check for possible similar corrections elsewhere).*

Corrected.

*4. P. 4, l. 12, ... it contains all the available information. That is vague. Can you be more precise?*

Corrected.

*5. P. 15, l. 7, a regularization parameter α, which appears near the term involving $Q$ and $Q(0)$ → ... parameter α, which weights the squared difference $|Q - Q(0)|^2$.*

Corrected.

*6. P. 14, l. 17, ... with zero initial condition ... I understand this means that the initial velocity was zero. But what about temperature ? What is uniform ?*

Zero initial conditions (including temperature, velocities, salinity) are taken to overclock the system, and this is done at the preliminary stage for the INM RAS model. Climatic data from the atmosphere are used as boundary conditions and partly in the right-hand side. Therefore, after 20 years of calculating the model, a solution close to the climate for the region is obtained. After this, the nonzero solution obtained is taken as the initial condition for the experiment described in this paper.

New explanations are introduced in the text of the revised version, and they are marked red.

We are greatly thankful to you for very useful critical remarks and comments which helped us to improve the paper.

Sincerely,

Victor Shutyaev, Francois-Xavier Le Dimet, and Eugene Parmuzin